# Strategically Altered Fluorinated Polymer at Nanoscale for Enhancing Proton Conduction and Power Generation from Salinity Gradient

**DOI:** 10.3390/membranes12040395

**Published:** 2022-04-01

**Authors:** Prem P. Sharma, Rahul Singh, Syed Abdullah Shah, Cheol Hun Yoo, Albert S. Lee, Daejoong Kim, Jeong-Geol Na, Jong Suk Lee

**Affiliations:** 1Department of Chemical and Biomolecular Engineering, Sogang University, 35, Baekbeom-ro, Mapo-gu, Seoul 04107, Korea; premsharma15@gmail.com (P.P.S.); dbcjfgns1234@naver.com (C.H.Y.); 2Department of Mechanical Engineering, Sogang University, 35, Baekbeom-ro, Mapo-gu, Seoul 04107, Korea; singhrs1195@gmail.com (R.S.); abdullahicp@gmail.com (S.A.S.); 3Korea Institute of Science and Technology, Hwarangno 14 gil-5, Seongbuk-gu, Seoul 02792, Korea; aslee@kist.re.kr

**Keywords:** ionic phase, semicrystalline, hydrophilic, salinity gradient, power density

## Abstract

Reverse electrodialysis (RED) generates power directly by transforming salinity gradient into electrical energy. The ion transport properties of the ion-exchange membranes need to be investigated deeply to improve the limiting efficiencies of the RED. The interaction between “counterions” and “ionic species” in the membrane requires a fundamental understanding of the phase separation process. Here, we report on sulfonated poly(vinylidene fluoride-co-hexafluoropropylene)/graphitic carbon nitride nanocomposites for RED application. We demonstrate that the rearrangement of the hydrophilic and hydrophobic domains in the semicrystalline polymer at a nanoscale level improves ion conduction. The rearrangement of the ionic species in polymer and “the functionalized nanosheet with ionic species” enhances the proton conduction in the hybrid membrane without a change in the structural integrity of the membrane. A detailed discussion has been provided on the membrane nanostructure, chemical configuration, structural robustness, surface morphology, and ion transport properties of the prepared hybrid membrane. Furthermore, the RED device was fabricated by combining synthesized cation exchange membrane with commercially available anion exchange membrane, NEOSEPTA, and a maximum power density of 0.2 W m^−2^ was successfully achieved under varying flow rates at the ambient condition.

## 1. Introduction

Power production from the salinity gradient of seawater and river water is considered a noise-free, clean, and sustainable route of energy-generating technology [1,2]. Theoretically, the global salinity gradient power (SGP) is estimated at around 1.4 to 2.6 TW, out of which ~60% energy can potentially be extracted (i.e., ~980 GW) [3]. The different methods used for producing SGP are reverse electrodialysis (RED) [4,5], capacitive mixing (CAPMIX) [6,7], vapor pressure difference utilization (VPD) [8], hydrocratic generation (HG) [9], and pressure-retarded osmosis (PRO) [10,11]. It has been shown that PRO and RED are two of the most promising technologies for producing power from a commercial point of view [12]. The RED technology offers enormous potential for producing efficient power at grid-level power plants [13,14]. The RED system has been investigated at a wide scale in research laboratories [15,16] as well as pilot-plant scale [17,18]. In order to make RED an efficient and competitive technology, several kinds of device engineering have been performed, such as RED with microbial fuel cells [19], RED with desalination process [20], RED with reverse osmosis (RO) process [20], RED with solar power [21], and RED with the radiative cooling process [22]. All these technological advances in the RED system make it an efficient and reliable energy source for overcoming energy crises. However, two main challenges restrict the commercialization of RED units: fouling and cost of the membrane [23,24,25]. Ion-exchange membrane fouling is primarily caused by multivalent ions, organic and inorganic particles, and natural microorganisms [23,26,27]. These foreign species directly affect the membrane’s physicochemical properties, resulting in an adverse effect on the membrane’s output performance in the long run. The RED fouling is fundamentally different from the electrodialysis (ED), and electrodialysis reversal (EDR) processes [23]. In general, the salinity gradient power generation was governed by monovalent ions. However, multivalent ions strongly contributed to the fouling formation of the IEMs. Several antifouling strategies have been introduced, including physical, chemical, and biochemical processes. The physical cleaning was carried out using reverse flow, removing air bubbles, ultrasonication, and mechanically are some popular approaches. In contrast, chemical cleaning was achieved by the use of chemical reagents, like acids (sulfuric), bases (soda), and oxidants (hydrogen peroxide). The biochemical cleaning process uses bioactive agents such as the enzymatic process to clean the membrane [28,29]. To restrain membrane fouling and cost issues, the easiest and most effective way is to identify new material or nanoengineer the polymer to develop a new low-cost, high-performance membrane for the RED system.

Nafion, Aquivion, Hyflon, and Selemion are commercially available perfluorosulfonic acid polymers (PFSA) and extensively used as ion-exchange membranes [30]. Although these perfluorinated polymers offer outstanding device performance and have shown enormous potential in various electrochemical devices [31], RED is composed of multiple alternate arrangements of cation-exchange membranes (CEMs) and anion-exchange membranes (AEMs). Thus, the use of these commercially available membranes significantly increases the overall cost of the system. To make the RED system commercially viable, a reduction in the cost of commercial membranes is essential. Poly(vinylidene fluoride)-co-hexafluoropropylene (PVDF-HFP) is a class of semicrystalline polymer. It offers remarkable physicochemical properties for developing free-standing, low-cost, high-performance ion-exchange membranes. It has already shown promise in water treatment, power generation, desalination, pharmaceutical, food processing, and gas separation applications [32]. PVDF-HFP exhibits excellent chemical, thermal, and structural stability under stress conditions. The most popular techniques for producing free-standing membrane are the non-solvent induced phase separation (NIPS) method [33] and the thermally induced phase separation (TIPS) method [34]. Typically, to develop an efficient cation-exchange membrane, the sulfonic acids (-SO_3_H) group is used widely due to its high and stable performance in the long run. In contrast, other popular functional groups are carboxylic acids (COOH), phosphonic acids (-PO_3_H_2_), and phenolic hydroxide due to the well-balanced amorphous and crystalline regions in PVDF-HFP polymer, a critical factor for determining the membrane’s high conduction and structural integrity.

Here in this work, we have demonstrated a technique for altering the hydrophilic region of the sulfonated-PVDF-HFP polymer at the nanoscale by using sulfonated-graphitic-carbon nitride nanosheets. Firstly, the hydrophobic polymer was dehydrofluorinated, then the sulfonation of the dehydrofluorinated fluorocopolymer was carried out. Simultaneously, the exfoliated nanosheet was prepared using the thermal oxidation etching technique followed by the sulfonation of the nanosheet. Finally, the prepared membrane was utilized as CEM for the fabrication of a reverse electrodialysis device. Apart from the fabrication of the device, the prepared membrane was subjected to an examination of its structural and chemical stability. Furthermore, in this article, we have demonstrated how crystallinity and amorphous regions play an essential role in governing membrane performance. The synthesized exfoliated nanosheets are confirmed through X-ray diffraction and HR-TEM results. High ion-exchange capacity (IEC) and enhanced proton conduction make sulfonated-PVDF-HFP membrane an appropriate choice for device implementation. Other essential parameters, such as water uptake, thermal degradation, and surface morphology, are also discussed in detail. We have shown that selecting an appropriate membrane combination can significantly enhance a device’s performance in the long run and reduce membrane fouling and cost.

## 2. Materials and Methods

### 2.1. Materials

Poly(vinylidene fluoride-co-hexafluoro propylene) were ordered in pellets from Sigma Aldrich (St. Louis, MO, USA). The dehydrofluorinated form of this polymer is termed DPVDF. Chlorosulfonic acid 98% pure was obtained from Kanto Chemical, Tokyo, Japan. *N*-methyl-2-pyrrolidone (NMP), dichloromethane, and dimethylacetamide (DMAc) were used as solvents and obtained from Daejung Chemicals, South Korea.

### 2.2. Synthesis of Dehydrofluorinated Polymer

The dehydrofluorination of the polymer was achieved by treating the fluoropolymer backbone with the saturated alkali solution [35]. Briefly, the required number of pellets was dissolved into DMAc at 50 °C until the solution became transparent and homogeneous. Subsequently, pellets of sodium hydroxide were also dissolved into iso-propyl alcohol (IPA) at room temperature. After the dissolution, NaOH solution was added dropwise to the homogeneous solution of the polymer. The transparent solution turned to a dark-colored solution. After that, this dark-colored solution was precipitated into DI water and repeatedly washed to remove any excess amount of base and dried in a vacuum oven at 80 °C overnight to obtain dehydrofluorinated product for further characterization.

### 2.3. Synthesis of Sulfonated Polymer

The dehydrofluorinated fluorocopolymer was dissolved in NMP. After its complete dissolution, the solution was cast on a clean glass plate and dried under vacuum at 70 °C for 18 h. The thickness was maintained at ~100 µm. After complete evaporation of the solvent, the membrane was peeled off and immersed in a 20 mL chlorosulfonic acid bath to get the sulfonated product. After this protocol, the membrane was slowly immersed into deionized water and repeatedly washed until reaching pH~7. The resultant membrane was the sulfonated product of the fluoropolymer.

### 2.4. Synthesis and Exfoliation of g-C_3_N_4_

Nanosheets were synthesized and exfoliated by an acid exfoliation thermal oxidation etching process. Briefly, 1 g of white melamine powder was first ultrasonicated and dispersed in 250 mL deionized water. Afterward, 15 mL of acid (sulphuric acid) was added dropwise into the solution mixture with vigorous stirring. The color of the solution turned from transparent to white color (indication of the endpoint). After that, the sample was kept in the freezer to lower the temperature of the solution mixture to minimize the heat produced by the neutralization reaction of concentrated sulphuric acid and water. The white powder form was then separated and transferred into an alumina crucible and subjected to a muffle furnace at 550 °C for 4 h with a heating and cooling rate of 5 °C/min. After that, the obtained yellow particles were milled into powder form with the help of a mortar. Milled powder was dispersed into water and centrifuged at 6000 rpm for 10 min, then the suspension was collected and dried to get exfoliated nanosheets.

### 2.5. Sulfonation of g-C_3_N_4_

These particles were sulfonated with chlorosulfonic acid to make functionalized two-dimensional nanosheets, as described earlier [36]. A total of 1 g of obtained particles were dissolved in 25 mL of dichloromethane. After that, 2 mL chlorosulfonic acid in 10 mL CH_2_Cl_2_ was added to this mixture dropwise until the color of substrate turned from yellow to white. The solution was then stirred for 3 h and centrifuged with deionized water to make it neutral, then freeze-dried to obtain a purified form of the particle.

### 2.6. Fabrication of Composite Membrane

The blending method was chosen for the fabrication of a composite cation exchange membrane. Different types of composite membranes were fabricated by solution casting method and designated as S-DPVDF, S-DPVDF-1, S-DPVDF-2, and S-DPVDF-5; 1, 2, and 5 represent the composition of sulfonated nanosheets (*w*/*w*%), i.e., sulfonated g-C_3_N_4_.

A predetermined quantity of sulfonated particles was briefly subjected to ultrasonication to obtain dispersed form in solvent NMP. Additionally, the desired amount of polymer was added to the same mixture with continuous stirring for 12 h. The homogeneous mixture was then cast on a clean glass plate, and thickness was maintained with the help of a casting knife. The membranes were dried in a vacuum oven overnight at 80 °C, then peeled off carefully and immersed into 1 M H_2_SO_4_ to complete ionization of sulfonic acid groups.

## 3. Results

### 3.1. Nanostructure of the Membrane

Nanostructure morphological changes originated in the membrane from the phase-separation process. It is a governing factor for providing outstanding physicochemical properties. A membrane’s phase separation is caused by two different regions: polymer high-concentration and low-concentration regions. Furthermore, these regions are categorized as nanochannel, nanopores, layered structures, and polymer bundles [37,38]. These shapes and sizes of the phase-separated domains depend on the hydrophobic and hydrophilic nature of the polymer. One can easily enhance the nanochannel size width by controlling the polymer low-concentration region or hydrophilic part in the membrane. Increasing nanochannel width to a certain level may deteriorate the structural integrity of the membrane. Therefore, it is recommended that polymer chain flexibility in the membrane plays a critical role while selecting a polymer. It is well known that chain flexibility depends mainly on the amorphous content of the polymer.

On the other hand, crystalline content provides structural robustness to the membrane. Thus, a semicrystalline polymer is a material that fulfills all fundamental requirements for developing a free-standing and high ion-conducting membrane. Polyvinylidene fluoride-co-hexafluoropropylene (PVDF-co-HFP) is considered a well-studied semicrystalline polymer in terms of morphological, structural, thermal, mechanical, and electrical properties [39,40,41,42]. However, a fundamental understanding of intrinsic structural properties on the transport phenomenon of the membrane is still lacking. The membrane nanostructure complexity can be understood by small-angle X-ray scattering measurement, which can easily determine ionomer structure with 1–100 nm size [43]. The presence of nanopores-or-nanochannel in the membrane resembles the ionomer peak at the “q” position, as illustrated in Figure 1a,b. Shifting of the ionomer peak at “q” position in a left or right direction can be attributed to a change in the polymer’s high concentration region and low concentration region.

The shifting of the ionomer peak position unveils the link between the change in the polymer concentration regions and semicrystalline behaviors. Usually, pure PVDF-HFP is composed of crystalline (or lamellae region), amorphous domains, and the intermediate link-region between the end of the lamellar phase and the start of the amorphous phase. This “intermediate link region (between crystalline and amorphous)” and the crystalline region mainly offer the rigid character, where only the amorphous part shows chain flexibility. Overall, pure PVDF-HFP membrane does not contain double bonds, and therefore offers high flexibility to the chain. Whereas before polymerization, hexafluoropropylene and vinylidene fluoride have double bonds making them very rigid, in pure PVDF-HFP membrane, the presence of ionomer peak at low “q” value ~0.075 ${Å}^{-1}$ can be attributed to the polymer matrix knee, representing intercrystallite domain spacing, as shown in Figure 1c. This peak relates to the PVDF-HFP lamellar structure and is only visible when the X-ray is placed far away from the charge-coupled detector (CCD) at 4 m. However, with sulfonated-PVDF-HFP measured at 1 m (distance between CCD and X-ray), a broad ionomer peak at 0.4 ${Å}^{-1}$ represents the ionic phase, which falls under the hydrophilic domain of the polymer with a less-concentration polymer region. Typically, these ionic groups (SO_3_H) are attached to the PVDF-HFP backbone (hydrophobic domain) through interconnected links (side chain), forming an ionic phase near the amorphous region [44]. The presence of the ionic phase is responsible for the proton conduction in the membrane; therefore, the higher the ionic phase, the better the conduction. It is important to note that if the ionic phase increases above the critical level, it can easily worsen the mechanical strength of the membrane. It is observed from the radius of gyration (R_g_) value of sulfonated-PVDF-HFP membrane that this membrane offers higher R_g_ values above 1.3 Å. It elucidates a well-distributed hydrophobic polymer backbone and ionic phases throughout the membrane, as presented in the histogram Figure 1d and Appendix A. The prepared sulfonated membrane shows ~1.6 nm wide ionic phase in the dry state of the membrane evaluated from Bragg’s equation, as tabulated in Appendix A. It is perceived from the broad ionomer peak of the ionic phase that there is an enhancement in the amorphous region, which leads to an increase in ion conduction. On the other hand, double log plots are divided into two areas, low-q side and high q-side, and the linear region of the plots represent the mass fractal dimension. From the power law ($I\left(q\right)\propto q^{-X_{m}}$), exponent ($-X_{m}$) is determined, and it appears that low q-side (region-1) shows ~3 in both membranes. The exponent value changes in the second region, i.e., high q-side, to 2.95 for the pristine membrane, and 3.6 for sulfonated-PVDF-HFP, as displayed in Appendix A. Obtained exponent values represent the mass fractal structural morphology and are evidence for the formation of an intra-group of interconnected branched nanostructure in the membrane.

### 3.2. Wide-Angle X-ray Diffraction

Both hydrophobic and hydrophilic domains are essential for the improvement of structural integrity and ionic conduction, respectively. Typically, PVDF-HFP exhibits α, β, γ,δ, and ε phases, which mainly arise from the regular arrangement of -CH_2_- and -CF_2_- groups in the polymer [45,46]. The β and γ crystallite phases offer dipole moments due to the presence of an electronegative fluorocarbon group, whereas other α,δ, and ε phases are polar in nature. Similarly, these crystalline phases are also observed in the dehydrofluorinated-PVDF-HFP sample and the sulfonated-PVDF-HFP sample, as shown in schematic Figure 2a. It is noted that at the low q-side of the sample, sulfonated-PVDF-HFP is unable to show a scattering peak in the SAXS plot. However, the presence of sharp crystalline peaks in all sulfonated-PVDF-HFP samples supports both SAXS and WAXD results. It is closely observed that the crystalline peaks (100, 110, 020, and 021) appear sharply on the broad-amorphous hump at 2θ value between 15–20, 25–30, and 35–45 degrees as illustrated in Figure 2b. The crystalline region in the membrane is mainly responsible for providing structural integrity. The ionic species attached to amorphous phases provide better ion conduction to the membrane, as illustrated in Figure 2c. It is noted that at the prominent crystallite peaks (at 2θ values 18.4, 26.4, and 38.8), relative intensity decreases as the ionic phase in the polymer increases.

On the other hand, no extra peaks appear after incorporating graphitic carbon nitride at 2θ = 27.4°, which confirms the well-distributed exfoliated nanosheet and no change in the chemical structure of the membrane. However, bulk graphitic carbon nitride particles show a sharp crystalline peak around 2θ = 27.4°. After exfoliation of the graphitic carbon nitride, a broad hump was replaced by the original peak, which can be attributed to the elimination of crystalline nature, as shown in Appendix A. The drastic decrease in the crystalline framework of the nanosheet signifies the disordered arrangement of the nanosheet. After incorporating functionalized graphitic carbon nitride, no significant transformation was observed in the polymer backbone. No alteration in the polymer semicrystalline phase signifies that the membrane maintains its structural integrity at the high water uptake and serves as the main constituent for membrane durability. On the other hand, the disordered arrangement of the exfoliated graphitic-carbon nitride nanosheet is confirmed from the scanning electron micrograph and the transmission electron micrograph, as illustrated in Appendix A. Elemental analysis was performed using energy-dispersive X-ray spectroscopy, which shows the carbon and nitrogen atomic weight % content. The mapping of the element confirms the uniform distribution of elements, as displayed in Appendix A. The structural modification of the graphitic carbon nitride was further confirmed from the FT-IR spectroscopy, as shown in Appendix A. A sharp peak in both bulk and exfoliated nanosheet at ~805 cm^−1^ wavenumbers confirms the presence of tris-triazene ring formation, whereas the primary and secondary amine group peaks appear near 3000 to 2800 cm^−1^ wavenumber, which shows a broad hump in this region that resembles -OH groups. It also signifies the presence of the SO_3_H group. The stretching vibration of -C=N- bond and -C-N- bond emerge around 1650–1568 cm^−1^ and 1426–1315–1220 cm^−1^ wavenumbers, respectively. Simultaneously, the sulfonation peak around 1195 and 1080 cm^−1^ represents the stretching vibration of the -S=O group.

### 3.3. Chemical Configuration of the Membrane

Chemical composition and identification of the functional group were confirmed from ^1^H-NMR spectroscopy measurement of the polymer. The ^19^F-NMR spectra of dehydrofluorinated-PVDF-HFP polymer exhibit multiplets of additional peaks with the reduction in the -CH_2_-CF_2_- peak position compared to pure PVDF-HFP. Other peaks in the multiplets correspond to (-CH_2_CH=CFCF_2_-) and (-CF_2_CH=CFCF_2_) around −111 to −112 ppm, as illustrated in Appendix A. That all these series of -C=C- based groups peak in ^19^F-NMR spectra confirms dehydrofluorination of PVDF-HFP polymer. Typically, the ^1^H-NMR spectra exhibit two characteristic multiplet peaks around 3.5 and 2.9 ppm due to the proton present on vinylidene fluoride unit (-CF_2_-CH_2_-CH_2_-CF_2_-) and (-CF_2_-CH_2_-CF_2_-CH_2_-), respectively, as presented in Appendix A. However, after dehydrofluorination of PVDF-HFP polymer, small peaks around 1.2 to 1.5 ppm represent the occurrence of a -C=C= bond [47]. After confirming the double bond from the ^19^F and ^1^H-NMR spectra, the structure of dehydrofluorination of PVDF-HFP polymer was treated with sulfonic acid. New signals appear around 6.75, and 6.73 ppm corresponds to the proton, which resembles the presence of sulfonated groups, as displayed clearly in the inset of Appendix A. The FT-IR spectra further verify the “head to tail” polymer’s arrangements plus the polymer sulfonation. It is observed that IR peak near fingerprint region 876 cm^−1^, 1194 cm^−1^, and 1390 cm^−1^ wavenumbers allocated to the stretching vibration of -CF_3_-, -CF_2_- and -CF- groups, respectively. However, a couple of tiny peaks that appear around 2900 to 2800 cm^−1^ signify the presence of symmetric and antisymmetric stretching vibration of -CH and -CH_2_ groups. It is noted that the characteristic peaks represent the -C=C- are not present in the pure PVDF-HFP of IR-spectra, as presented in schematic diagram Figure 3a. The appearance of a peak at ~1646 cm^−1^ wavenumber for the dehydrofluorinated-PVDF-HFP sample is assigned to the -C=C- bond. The presence of a double bond confirms the dehydrofluorination reaction and further verifies the NMR data. The formation of double bonds is further processed through the sulfonation reaction. A new peak arises at ~1029 cm^−1^ wavenumber, ascribed to the symmetric and asymmetric stretching vibration of -S=O- bond. It confirms the presence of the -SO_3_H group in the sulfonated-PVDF-HFP polymer, as shown in Figure 3b.

### 3.4. Structural Robustness

The prepared sulfonated membrane shows stable performance and maintains its free-standing structure under-hydrated conditions. The prepared membrane’s hydrophilicity content is estimated to check the membrane’s wettability (see Figure 4a). It is observed that the adhesion of water droplets on the surface of the pure PVDF-HFP membrane offers low hydrophilic nature and the calculated contact angle ~84°. However, the hydrophilicity significantly increases in the sulfonated-PVDF-HFP based membrane. The contact angle of the water droplet on the surface of the sulfonated membrane is ~77°. This significant drop in the value of contact angle from 84° to 77° is due to the interaction of the sulfonic group with the H_2_O molecule, which was explained earlier (see Figure 2c).

Moreover, we observed the behaviors of the contact angle with the time duration of 10 min, as displayed in Figure 4b. It is observed that as time increases, the contact angle further drops down from 77° to 67° without deteriorating the membrane nanostructure. This drastic drop in the contact angle signifies that the adhesion of water molecules to the surface of the membrane is excellent. Further, this membrane can be utilized in electrochemical devices. The thermal robustness of the prepared membrane at elevated temperature was determined by TGA analysis, as illustrated in Figure 4c. It is observed that the first weight loss occurs around 150 °C, which is due to the evaporation of the absorbed moisture by membranes. The second weight loss is observed around 270 to 360 °C. This weight loss is caused due to sulfonic acid degradation. The desulfonation of polymer is the critical degradation that collapses the ionic species of the membrane. Thus, the prepared sulfonated-PVDF-HFP membrane is thermally stable up to 150 °C. As the content of exfoliated nano-sheets increases, the thermal stability of the membrane enhances slightly.

Furthermore, membranes start decomposing when the temperature increases above 400 °C. A significant drop in the weight of the membrane around 500 °C is attributed to backbone chain degradation. The complete oxidation of polymer backbone -CH_2_-CF_2_- occurs around 500 °C to 600 °C, which shows massive weight loss. This weight loss reveals the strong inter-ionic hydrogen bonding interaction between the sulfonated graphitic carbon nitride substituted as nanofiller and sulfonated polymer, which upsurges the composite membrane’s thermal stability, as shown in Figure 4c. Excellent compatibility of the nanosheet for the composite membrane positively alters the thermal stability. Additionally, the amount of nanosheets incorporated inside the sulfonated polymer was also measured by measuring the residual weight after complete combustion of polymer backbone in the oxygen environment through TGA analysis. It was calculated as 0.64, 1.23, and 3.49 wt% for the samples S-DPVDF-1, -2, and -5, respectively.

Typically, as the concentration of the sulfonic acid group increases in the membrane, it is noted that the water uptake increases drastically, which can quickly worsen the structural integrity of the membrane. Despite the increase in the ion exchange capacity of the prepared membrane, a slight increase in the water uptake is observed from 9.9 to 19.5%, as tabulated in Table 1 and Appendix A. The λ(H_2_O/SO_3_H) of prepared CEM governs the water uptake capacity and is directly proportional to the membrane’s ion conduction. Generally, the bound water region in the membrane represents the region where the H_3_O^+^ ions surround the ${\mathrm{SO}}_{3}^{-}$ group. Usually, when an excess H_2_O molecule surrounds the SO_3_H group, a free-volume space is created, resulting in membrane swelling. This increase in the water uptake confirms the increase in the hydrophilic nature of the membrane, as it was observed from the contact angle measurement (see Appendix A). This increase in water content in the membrane provides a large number of H_2_O molecules per sulfonic acid group, resulting in better proton conduction. The vehicle mechanism is responsible for the proton conduction in the ion-exchange membrane, which promotes H_3_O^+^ ions [48]. High ionic conductivity is a fundamental requirement for delivering efficiency and high performance in any membrane in an electrochemical device. It is noted that as the IEC value of the membrane increases, the proton conductivity also increases significantly from 1.15 × 10^−3^ to 3.6 × 10^−3^ S cm^−1^. This improvement in the proton conductivity signifies the expansion of the hydrophilic nanochannel width in the membrane. This enhancement of the hydrophilic domain, as previously observed in the SAXS result, resembles the presence of excess free water. A comparative Appendix A represents the commercially available perfluorosulfonic acid membrane.

### 3.5. Surface Morphology of the Membrane

Membrane morphology mainly depends on the phase separation process, which determines the uniform distribution of crystalline and amorphous regions. FE-SEM is used to analyze the surface morphology of the membrane at a 2-micrometer scale, as presented in Figure 5a,b. It displays that the pure PVDF-HFP membrane shows a dense morphology where no porous morphology is observed on the surface of the pure PVDF-HFP membrane, and this is due to the formation of a dense polymer network. However, sulfonated-PVDF-HFP with exfoliated nanosheet-based membrane shows a wrinkled morphology without obvious agglomeration, signifying a well-distributed nature of nanosheets throughout the membrane. The membrane surface roughness was further confirmed from the atomic force microscopy, as illustrated in Figure 5c. The surface topography demonstrates a defect-free uniform dense membrane. It implies that the prepared altered sulfonated-PVDF-HFP membrane shows a nanometer size phase-separated region, which could not be visible through this technique. Thus, the calculated value ~1.6 nm wide ionic phase in the dry state of the membrane evaluated from Bragg’s equation using small-angle X-ray scattering is relevant. The top surface of the modified sulfonated-PVDF-HFP membrane shows roughness at the nanometer scale. The estimated average root means square of high deviation R_q_ value at different regions was recorded, as presented in Figure 5d. Topographic roughness was calculated using horizontal and vertical lines, and the computed R_q_ value lies between 36 to 40 nm for horizontal lines and 3.5 to 7.5 nm for vertical lines. Both techniques confirm the low roughness of the membrane. The membranes’ smooth surface provides an efficient interaction with saltwater when implemented in the RED device.

### 3.6. RED Cell Performance

The maximum power density for the reverse electrodialysis experiment could be attained at the point where the resistance of external load becomes equal to the internal resistance of the RED stack [49]. Internal resistance is composed of three main parts: boundary layer, bulk layer, and ohmic resistance [50]. During the RED experiment, these values should be minimized up to an extent to attain maximum power output. Ohmic resistance is predominantly influenced by the nature and property of the material used. In our case, we obtained a maximum power density value which is 0.20 Wm^−2^ for S-DPVDF-5 (CEM), when combined with Neosepta (AEM) at a flow rate of 100 mL min^−1^. It is comparable to the power density value for a pair of commercially available membrane Neosepta (AEM and CEM) at the same flow rate (see Figure 6a and see Appendix A). It implies the synergistic effect arises due to the strong hydrogen bonding between sulfonated groups of the nanosheets with polymer. The fully dispersed and exfoliated nano-sized nanosheets embedded into the polymer matrix improve the hopping ionic transport by exposing maximum acidic sites. On the other hand, an increment in the hydrophilicity of sulfonated membranes will enhance the diffusion transportation of ions. Higher transport of ions will lower the ohmic resistance, which upturns the power density value.

The second and most important factor is the bulk and boundary layer resistance by which the power generation value in the RED experiment is greatly affected. These are included in the non-ohmic resistance of the system. The influence of non-ohmic resistance can be limited by exposing the effective area of the membrane with the electrolyte solution at a fast rate (see Figure 6b). It displays the variation in power density value with the flow rate. Initially, at a higher flow rate, we attain maximum power density value for our synthesized altered cation exchange membrane because voltage value is significantly increased to produce high open-circuit voltage. This open-circuit voltage decreases considerably at a lower flow rate because of concentration polarization, which can be diminished by varying the flow rate of solutions [51]. This study also implies the dependence of the stack’s performance on hydrodynamic conditions.

Moreover, the interface created between the membrane-solution and solution-spacers lowers the actual concentration gradient. Thus, it becomes an obstacle for ion transportation at a lower flow rate. An effective increment in the linear flow rate will enhance the concentration gradient by lowering the concentration polarization at the membrane, solution, and spacer interfaces. On the other hand, the impact of ion transport becomes more apparent at the minimum flow rates that may indicate the significant drop of electromotive forces. At relatively higher flow rates, the change in open circuit voltage values for different membranes is generally not noticeable because of a sudden decrease in the concentrations of ions with their respective compartments. From the above investigation, we can observe that the insertion of sulfonated graphitic carbon nitride into sulfonated dehydrofluorinated polymer enhances the ionic conductivity and increases the hydrophilicity of the membrane by maintaining structural durability. Moreover, it is observed that the uniform distribution of nano-sized exfoliated nanosheets with the highest concentration is more beneficial in terms of intermolecular reaction. Consequently, the prepared hybrid membrane acts as a promising candidate for next-generation RED systems (see Figure 6c). Comparative RED device performance of the commercially available membranes has been shown in Appendix A.

## 4. Conclusions

Excess-free water molecules that interact with the sulfonic acid group are responsible for the enhancement in the proton conduction in the membrane. However, excess water uptake by the membrane may deteriorate the structural integrity of the membrane. This article demonstrates that the crystalline and amorphous domain in the modified PVDF-HFP membrane provides structural integrity and better conduction. This work aims to develop a low-cost, free-standing high-performance membrane for reverse electrodialysis application. Furthermore, the altered ion exchange membrane with varying functionalized nanosheets enhances the membrane’s physicochemical properties. Physicochemical properties of the membrane nanostructure were discussed in detail using SAXS and WAXD analysis. The prepared membrane offers improvement in the ion exchange capacity without worsening its structure, resulting in an enhancement in the proton conduction up to 3.64 × 10^−3^ S cm^−1^. The obtained cell power density is ~0.2 Wm^−2^ when the sulfonated PVDF-HFP membrane is used as CEM and Neosepta as AEM. The power output performance measured by using a pair of the prepared S-DPVDF membrane and the commercially available Neosepta AEM membrane exhibits stable performance with a slight increment ±0.015 Wm^−2^ at varying flow rates from 60 to 100 mL min^−1^ without damaging any membrane integrity.

## Figures and Tables

**Figure 1 membranes-12-00395-f001:**
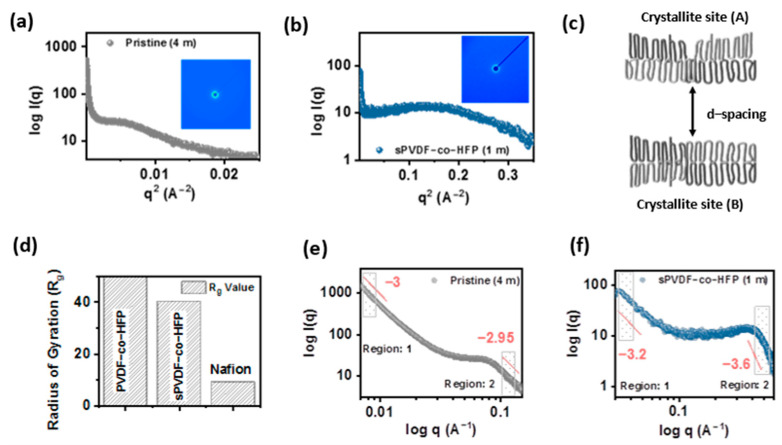
Small Angle X-ray scattering of the prepared membrane: (**a**) Pristine PVDF–co–HFP membrane with charge–coupled device image fixed at 4 m, (**b**) sulfonated–PVDF–co–HFP membrane with charge–coupled device image fixed at 1 m, (**c**) d–spacing, (**d**) histogram of the radius of gyration (R_g_), (**e**) double log plot of the pristine membrane, and (**f**) double log plot of the sulfonated–PVDF–co–HFP membrane.

**Figure 2 membranes-12-00395-f002:**
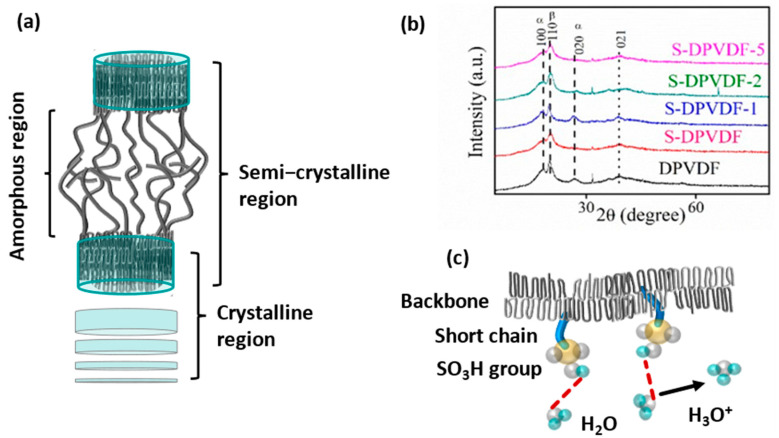
(**a**) Schematic representation of the semicrystalline network in PVDF–HFP membrane, (**b**) wide-angle X-ray diffraction of the prepared dehydrofluorinated and sulfonated–PVDF–HFP polymer, and (**c**) schematic representation of the interaction between H_2_O molecule with SO_3_H functional group with the formation of hydronium H_3_O^+^ ion.

**Figure 3 membranes-12-00395-f003:**
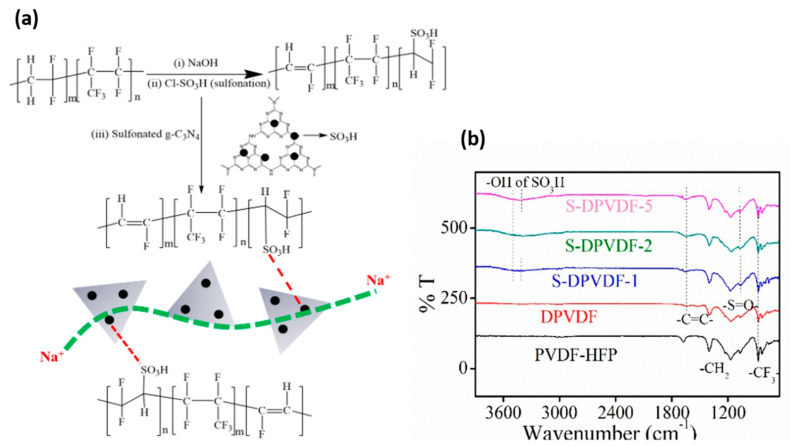
(**a**) Schematic diagram dehydrofluorination of PVDF–HFP and sulfonation of the PVDF–HFP polymer, and (**b**) Fourier–Transform Infrared spectroscopy of the pure PVDF–HFP, dehydrofluorination of PVDF–HFP, and sulfonation of the PVDF-HFP polymer with exfoliated nanosheet.

**Figure 4 membranes-12-00395-f004:**
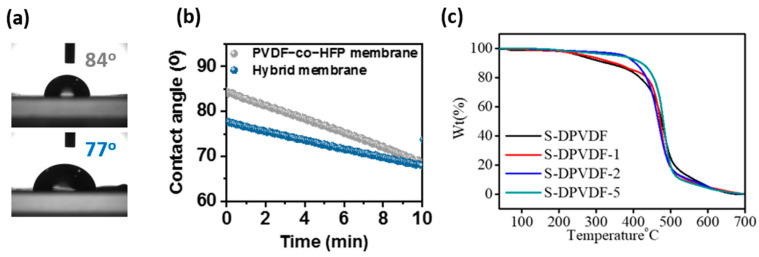
(**a**) Photograph of fresh water-droplet on the surface of pure PVDF–HFP and sulfonated–PVDF surface, (**b**) contact angle measurement of the pure PVDF and sulfonated PVDF–HFP modified with exfoliated nanosheet, and (**c**) thermal analysis of sulfonated–PVDF–HFP with varying amount of exfoliated nanosheet.

**Figure 5 membranes-12-00395-f005:**
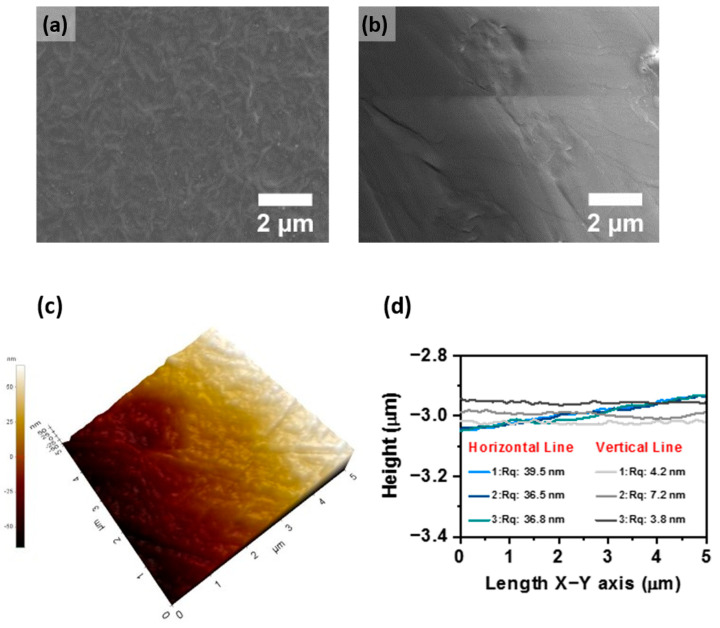
Surface morphology of the membrane: (**a**) FE-SEM of poly(vinylidene fluoride)-co-hexafluoropropylene (PVDF-HFP) modified with exfoliated nanosheet, (**b**) FE-SEM of sulfonated-PVDF-HFP, (**c**) AFM of sulfonated-PVDF-HFP modified with exfoliated nanosheet, and (**d**) surface roughness of sulfonated-PVDF-HFP modified with exfoliated nanosheet.

**Figure 6 membranes-12-00395-f006:**
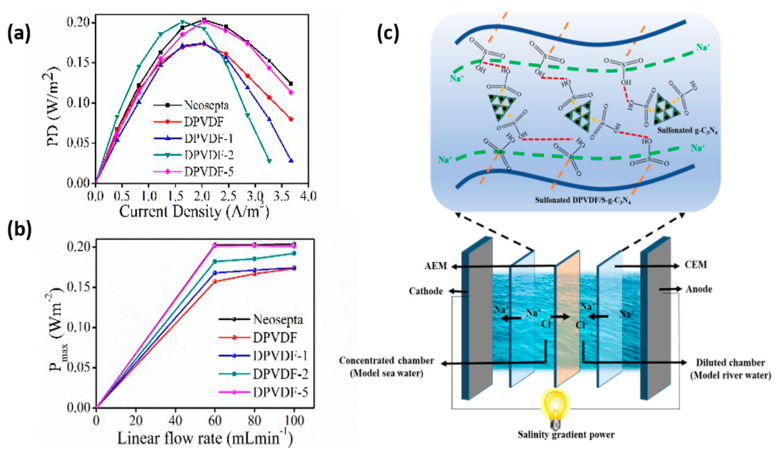
Reverse electrodialysis output performance: (**a**) Power density of prepared membrane, (**b**) maximum power with varying flow rate, and (**c**) schematic diagram of the reverse electrodialysis setup and mechanism representing formation of ionic cluster enhancing proton conduction.

**Table 1 membranes-12-00395-t001:** Electrochemical and physicochemical parameters of membrane ion-exchange capacity (IEC), water uptake (WU), and ion-conductivity (IC) of the different composite membranes.

Membrane Type	IEC (meq g^−1^)	Free Water (%)	Bound Water (%)	Water Uptake (%)	IC S cm^−1^
**S-DPVDF**	0.21	9.82	0.12	09.94	1.15 × 10^−3^
**S-DPVDF-1**	0.24	12.34	0.16	12.50	1.36 × 10^−3^
**S-DPVDF-2**	0.26	14.81	0.40	15.21	2.23 × 10^−3^
**S-DPVDF-5**	0.30	19.15	0.42	19.57	3.64 × 10^−3^

## Data Availability

Not applicable.

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
