# Peer review of "Strategically Altered Fluorinated Polymer at Nanoscale for Enhancing Proton Conduction and Power Generation from Salinity Gradient"

_membranes, 2022, doi:10.3390/membranes12040395_

Round 1

Reviewer 1 Report

The aim of this manuscript is relevant and useful in connection with the need to develop new membrane materials with low cost and high conductivity as well as to improve energy-generating technology. However, there are some comments on the manuscript.

1) It is necessary to add a comparison of the physicochemical properties of the membranes obtained in this work with commercial membranes.

2) The terms “Free water” and “Bound water” are not quite clear (line 411; line 402-403). It is not obvious why authors attributed the increase in “free water” to the SAXS results (line 402-403). Section 3.2. Wide-angle X-ray diffraction does not contain any reasoning about the water content.

3) What mechanism of fouling was assumed in sodium chloride solutions, which the authors used to study reverse electrodialysis? The conclusion about the absence of fouling in the process of editing is not general and can only be attributed to those conditions that were used in this work. This part of the conclusion needs to be corrected.

Misprints:

Line 411 (Table 1):

(mmol-eq g-1) instead of (meqgm-1). According to IUPAC recommendation [Nomenclature for chromatography (IUPAC Recommendations 1993) L. S. Ettre // Pure &Appl, Chem., Vol. 65, No. 4, pp. 81H72, 1993. https://doi.org/10.1351/pac199365040819 ], the unit of capacity is amount (mmol) of ionogenic group per mass (g) of dry ion exchanger.

IC x 103 instead of IC x 10-3 (If IC x 10-3 is equal to 1,15 S cm-1 than IC is equal to 1150 S cm-1, and If IC x 103 is equal to 1,15 S cm-1 than IC is equal to 0.00115 S cm-1).

Author Response

Reviewer 1

The aim of this manuscript is relevant and useful in connection with the need to develop new membrane materials with low cost and high conductivity as well as to improve energy-generating technology. However, there are some comments on the manuscript.

1) It is necessary to add a comparison of the physicochemical properties of the membranes obtained in this work with commercial membranes.

Response: We appreciate the reviewer’ suggestion. We have summarized all the physicochemical properties and ion exchange capacity for the commercial membranes in Table S3 for a comparison with those of the membrane developed in this work.

Table S3. Summary of physicochemical properties for commercial membranes and the s-PVDF developed in this work.

Specification

s-PVDF

(This work)

Aquivion[1]

Solvay

Nafion[2] N115 DuPont

Nafion[2]

N117 DuPont

Hyflon [3] Solvay

Neosepta[4] CSE

ASTOM

Selemion[5] CMV

Asahi

Water uptake (%)

19.57

~35

38

38

45

25

22

Ion exchange capacity (meq g-1)

0.30

1.02

0.95-1.01

0.95-1.01

0.825

1.5-1.8

1.3-2.4

Conductivity

(S cm-1)

3.64x10-3 at 23 oC

0.02

at 50 oC, 50% RH

0.1

at 23 oC, 50% RH

0.1

at 23 oC, 50% RH

~0.1

1.8 (ohm cm2)

 at 25oC NaCl sol

~0.006 in NH4Cl sol

Thickness (m)

~100

24.6 ±â€¯0.7

127

183

63

160

137

Solvent

N-methyl-2-pyrrolidone (NMP)

Isopropanol (IPA)

Isopropanol (IPA)

Isopropanol (IPA)

-

-

-

2) The terms "Free water" and "Bound water" are not quite clear (line 411; line 402-403). It is not obvious why authors attributed the increase in "free water" to the SAXS results (line 402-403). Section 3.2. Wide-angle X-ray diffraction does not contain any reasoning about the water content.

Response: Typically, the water capacity absorbed by the membrane is directly proportional to the membrane's ion conduction. It is because water present inside the membrane provides a medium for the transport of ions. Therefore, the  (H2O/SO3H) plays a vital role in determining the membrane performance. When the  group is surrounded by H3O+ ions, it represents the bound water region. However, the subtraction of bound water from water uptake value gives us free-water content. This free water region is responsible for the excessive swelling of the membrane. The membrane excess water uptake creates free volume around the hydrophilic region. Usually, when an excess H2O molecule surrounds the SO3H group, a free-volume space starts to expand, and thereby swelling the membrane. In contrast, the hybrid membrane exhibits a high resistance against swelling. Generally, high swelling is associated with the membrane with a high IEC value. Several approaches have been adopted to overcome the high swelling issue, including tuning the IEC value, long-chain functional group, and incorporating organic and inorganic particles. The use of metal-free functionalized g-C3N4 with semicrystalline polymer attempted in the current work is a unique approach because of the ionic interaction offered by the functional groups present in the polymer as well as functionalized g-C3N4.

From the SAXS result, it is estimated that the sulfonated membrane shows ~1.6 nm wide ionic phase in the dry state of the membrane (Page 5; line 222). This ionic phase in the polymer is responsible for the water uptake in the membrane and also responsible for the proton conduction in the membrane. Therefore, the higher the fraction of an ionic phase, the greater the water uptake and the better the ion conduction. (Now, we have modified the statement in the main manuscript. Please see page 9 marked with yellow color)

Changes made in the main manuscripts (Please see page no. 9 marked with yellow color):

The  (H2O/SO3H) of the prepared CEM governs the water uptake capacity and is directly proportional to the membrane's ion conduction. Generally, the bound water region in the membrane represents the region where the H3O+ ions surround the  group. Usually, when an excess H2O molecule surrounds the SO3H group, a free-volume space is created, adversely affecting the membrane swelling.

Section 3.2. Wide-angle X-ray diffraction does not contain any reasoning about the water content.

Regarding Wide-angle X-ray diffraction

We have used the wide-angle X-ray diffraction measurement to identify the polymer's semicrystalline region. The crystallinity of the membranes is confirmed through the wide-angle X-ray diffraction, which exhibits a peak at 2=~18.4, 26.4, and 38.8 o. These crystalline peaks represent the deformation of the crystalline structure due to the introduction of the functional group in the form of -SO3H, which resembles the formation of the covalently bonded polymer backbone with the side chain.

Points to be noted from the Wide-angle X-ray diffraction plots:

Firstly, the pristine PVDF-HFP polymer shows a high crystalline peak in the X-ray diffraction pattern owing to the intrinsic crystalline nature.

Secondly, sulfonated-PVDF-HFP shows a decrease in relative crystalline peaks or sharp crystalline domains because the presence of sulfonic acid groups interferes with the regular arrangement of polymer chains.

Third, the sulfonated PVDF-HFP/g-C3N4 hybrid membrane seems to show no presence of impurity or change in the polymer backbone after incorporating g-C3N4 particles.

It is required to have a semicrystalline phase that gives structural integrity to the membrane at the high-swelling state and acts as the main constituent for durability. Note: there is no significant movement in the peak position of the sulfonated-PVDF-HFP? in dry conditions.

Changes made in the main manuscript (Please see page no. 6 marked with yellow color):

After incorporating functionalized g-C3N4, no significant transformation was observed in the polymer backbone. No alteration in the polymer semicrystalline phase signifies that the membrane maintains its structural integrity at the high water uptake and serves as the main constituent for membrane durability.

3) What mechanism of fouling was assumed in sodium chloride solutions, which the authors used to study reverse electrodialysis? The conclusion about the absence of fouling in the process of editing is not general and can only be attributed to those conditions that were used in this work. This part of the conclusion needs to be corrected.

Response:  We thank the reviewer for the suggestion. We fully understand the reviewer's concern regarding the fouling, but unfortunately, we did not attempt to carry out the fouling measurement in the current work. However, the hybrid membranes were reported to show an excellent anti-fouling behavior [6, 7] and we expect that our sulfonated PVDF-HFP/g-C3N4 hybrid membrane would show a good long-term performance.

Changes made in the main manuscript (Please see page no. 13 marked with yellow color):

The power output performance measured by using a pair of the prepared S-DPVDF membrane and the commercially available Neosepta AEM exhibits stable performance with a slight increment 0.015 Wm-2 at varying flow rates from 60 to 100 mL min-1 without damaging any membrane integrity.  

Misprints:

Line 411 (Table 1): (mmol-eq g-1) instead of (meqgm-1). According to IUPAC recommendation [Nomenclature for chromatography (IUPAC Recommendations 1993) L. S. Ettre // Pure &Appl, Chem., Vol. 65, No. 4, pp. 81H72, 1993. https://doi.org/10.1351/pac199365040819 ], the unit of capacity is amount (mmol) of ionogenic group per mass (g) of dry ion exchanger.

IC x 103 instead of IC x 10-3 (If IC x 10-3 is equal to 1,15 S cm-1 than IC is equal to 1150 S cm-1, and If IC x 103 is equal to 1,15 S cm-1 than IC is equal to 0.00115 S cm-1).

Response: We apologize for all the incorrect notations and typos. We marked all the corrections with a yellow color. Please see the revised manuscript. (Page no. 10).

Changes made in the main manuscript:

Membrane type

IEC

(meq g-1)

Free water

(%)

Bound water

(%)

Water uptake (%)

IC

(S cm-1)

S-DPVDF

0.21

9.82

0.12

09.94

1.15x10-3

S-DPVDF-1

0.24

12.34

0.16

12.50

1.36x10-3

S-DPVDF-2

0.26

14.81

0.40

15.21

2.23x10-3

S-DPVDF-5

0.30

19.15

0.42

19.57

3.64x10-3

Reviewer 2 Report

The authors report the preparation, characterization, and application of an altered fluorinated polymer-based nanocomposite as cation exchange membrane for RED application. A high number of characterization techniques are applied, even though the discussion of the results must be improved before publication. Besides, the comparison with the literature in terms of power density should be included. The reviewer comments can be seen below:

- Since fouling usually has a major effect on AEMs than on CEMs due to the negative charge of natural organic matter (such as humic acids in river water streams), the reviewer wonders why the authors focused on developing polymers for improved proton conduction in CEMs. Apparently, the obtainable net power density by RED under natural streams is more affected by the effect of fouling on AEMs in comparison with the fouling effect on CEMs. The authors should extend the background on fouling concerns in the introduction section.

- The authors also focused on chemical modification as a promising antifouling strategy. However, there are alternative physical or enzymatic methods that could also be considered. Besides, membrane cleaning also represents an useful technique to increase the lifetime of the membranes by reducing surface fouling. The reviewer would like to recommend the authors to include these topics in the discussion of the manuscript. 

- During the fabrication of the composite membrane, environmentally unfriendly solvents such as N-methyl-2-pyrrolidone (NMP), dichloromethane and dimethylacetamide (DMAc) are utilized. In the last years, the research community is seeking for developing more environmentally friendly polymers for membrane surface modification and membrane fabrication for both CEMs (10.1016/j.desal.2019.114183) and AEMs (10.3390/membranes10060134). Could the authors provide a greener alternative for future research to substitute the above-mentioned solvents or propose a more environmentally-friendly synthesis route?

- Since the contact angle of the prepared membrane decreased (higher hydrophilicity) after 10 min, the reviewer wonders if the authors perform any characterization after using the membrane in the RED stack. What is the final contact angle value after use? The same is applicable for the proton conductivity of the membranes and other physico-chemical properties. This will provide useful information to improve the discussion of this work.

- The reviewer would like to know if the authors determine the membrane electro-resistance, which represents an essential figure of merit in RED approaches.

- What is the experimental time in the RED stack assembly? How is the evolution of the power density with time for the maximum value?

- The model solutions that the authors utilized in the RED stack were based on NaCl (0.6 M and 0.01 M, respectively). However, it is widely known the high negative effect of multivalent ions (present in natural streams) on the generated power density (please see as relevant references: 10.1016/j.memsci.2008.12.042; 10.1016/j.memsci.2019.117385). To improve the quality of the work and to show the behavior of the prepared membranes in a more real scenario, the reviewer would like to recommend the authors to introduce representative model solutions (e.g., MgCl2, CaCl2 and Na2SO4) in the system.

- The authors report a maximum power density of 0.2 W/m2 using S-DPVDF-5 (as CEM) and a commercial Neosepta AEM at a flow rate of 100 ml/min. However, the same result is achieved when using both commercial Neosepta CEMs and AEMs. In other words, the use of the prepared CEM do not improve the performance of commercial pristine ion-exchange membranes regardless the higher proton conductive properties of the prepared composites. The authors should make a comprehensive comparison between the commercial CEM and the prepared sample (stability, physico-chemical properties, proton conduction, electro-resistance, etc.). The authors must also provide future prospects for improving the performance of the prepared membranes in the RED system (membrane preparation, surface modification, etc.). What is the main reason to achieve the same performance in comparison with the use of the commercial Neosepta CEM? The discussion in this respect must be improved.

- The comparison with the literature should be included to highlight the potential applicability of the novel composites proposed. 

- The authors conclude that the prepared membrane shows stable performance for RED without fouling. However, any model foulant is added into the model solutions in the experiments with the RED stack (only NaCl aqueous solutions are involved). This should be clarified and justified. The conclusions must be improved, highlighting the improved characteristics of prepared samples.

Author Response

Reviewer 2

The authors report the preparation, characterization, and application of an altered fluorinated polymer-based nanocomposite as cation exchange membrane for RED application. A high number of characterization techniques are applied, even though the discussion of the results must be improved before publication. Besides, the comparison with the literature in terms of power density should be included. The reviewer comments can be seen below:

Response:  We appreciate the reviewer’s positive comments. Here we would like to mention that RED performance mainly depends on the combination of cation-exchange membrane and anion-exchange membrane. Suppose CEM has high ionic conductivity and AEM has relatively low conductivity. Then AEM will be considered as the governing membrane for RED performance. Thus, a direct comparison of efficiency with other reported literature may be vague or misleading in this case. Most of the commercially available membranes including Neosepta ACS/CMS (Tokuyama), FumasTech FAD/FKD (GmbH), Qianqiu AEM/CEM, (Hangzhou Qianqiu), Selemion AMV/CMV (Asahi Glass), Fuji AEM/CEM (Fujifilm), PC-SK and PC-SA (PCCell), and Ralex CMH- PES/AMH-PES (MEGA) offer maximum power ranging from 0.33 to above 1 W m-2 (see table). However, a handful of literature is available, which talk about handmade or tailor-made membrane for reverse electrodialysis. A combination of PECH and sPEEK_1.76 based membrane[8] offers a power density of 1.28 W m-2 at room temperature with a swelling density of 49 % and 35.6 %, respectively. The power density of s-PVDF (hybrid membrane) and Neosepta based membrane offers 0.2 W m-2 at ambient conditions, with a stable performance at varying flow rates and negligible swelling density.

Changes made in the supporting information section of the manuscript are present below:

Table S4. Summary of the power density for commercially available membranesand the sPVDF/Neosepta membrane used in this work. Here, A: Active membrane area; N: Number of cell pairs; Pmax : Maximum power density.

Type

Area (cm2)

N

Pmax (W/m2)

Ref

sPVDF/Neosepta

7x7

1

0.2

This work

Neosepta CMX (Tokuyama)/Fuji T1 AEM (Fujifilm)

10x10

5

~0.2*

[9]

Ralex CMH/AMH, (MEGA)

10x10

5

∼0.26* 

[10]

Neosepta AMX/CMX (Tokuyama)

10x10

10

~0.42*

[10]

Fuji T1 CEM/Fuji T1 AEM (Fujifilm)

6.5x6.5

10

~0.7*

[11]

Neosepta ACS/CMS (Tokuyama)

10x10

5

3.8

[12]

FumasTech FAD/FKD (GmbH)

10x10

50

6.7 (60 °C)

[12]

Qianqiu AEM/CEM, (Hangzhou Qianqiu)

25x75

25

∼0.83

[13]

Selemion AMV/CMV (Asahi Glass)

10x10

5

1.18

[14]

FumaTep FKD and FAD (GmbH)

10x10

25

1.17

[14]

FumaTep FKD and FAD (GmbH)

10x10

50

0.93

[15]

Qianqiu Heterogeneous AEM/CEM (Hangzhou QianQiu)

10x10

5

1.05

[14]

Fuji AEM/CEM (Fujifilm)

10x10

25

1.06

[16]

PC-SK and PC-SA (PCCell, Germany)

10x10

10

~0.33

[17]

Neosepta AMX/CMX (Tokuyama)

100

3

0.87

[18]

Neosepta CMX/AMX (Tokuyama

6x13

5

0.59

[19]

FKS/FAS (FumaTech GmbH)

10x10

5

2.2

[20]

FKS/FAS (FumaTech GmbH)

10x10

5

0.5

[20]

Neosepta CMX/AMX (Tokuyama)

10x10

2-30

0.95

[21]

Ralex CMH- PES/AMH-PES (MEGA)

5x5

5

0.62

[22]

Ralex CMH- PES/AMH-PES (MEGA)

10x10

5

0.65

[22]

- Since fouling usually has a major effect on AEMs than on CEMs due to the negative charge of natural organic matter (such as humic acids in river water streams), the reviewer wonders why the authors focused on developing polymers for improved proton conduction in CEMs. Apparently, the obtainable net power density by RED under natural streams is more affected by the effect of fouling on AEMs in comparison with the fouling effect on CEMs. The authors should extend the background on fouling concerns in the introduction section.

Response:  We appreciate the reviewer’s valuable comments! We agree that the membrane fouling is a critical factor of regulating the stack's power output. Fouling of RED or membrane or spacer or compartment is an extensive topic. We did not attempt to cover fouling specifically here because our prime objective deals with polymer chemistry and is to prepare a low-cost hybrid membrane for power production.

Generally, the primary contributor to membrane fouling is organic matters, inorganic materials, colloids, microbial byproducts, etc. Please note: the fouling behavior in electrodialysis (ED) and electrodialysis reversal (EDR) differs from that in RED. Most of the foulant has previously been investigated for ED, especially AEM. Clearly, fouling in RED and the specific role of each membrane is not fully understood yet and is still an open topic for debate.

Although several attempts have been made previously, the salinity gradient power generation was generally governed by monovalent ions. However, multivalent ions strongly contributed to the fouling formation of the ion-exchange membranes (IEMs). We did not use multivalent ions for salinity gradient power generation in our case. Here, we have only used laboratory-grade NaCl salt solution for salinity gradient power generation.

Mainly, in this work, fouling is attributed to the running membrane in the stack at the varying flow rate of the feed solution. Here, the effect of fouling on RED is compared at a single membrane pair stack (power output) at a varying flow rate to determine the overall RED performance. The RED power output enhancement at higher flow rates is mainly due to decreased diffusion boundary layer resistance, which the fouled stack could not observe.

On the other hand, finding the appropriate fouling type is challenging and require extreme experimental conditions. That can divert the scope of this work. Therefore, we focused on the single controlled type at varying flow rates to check the limiting the performance of RED, rather than focusing on different types.

Changes made in the main manuscript (Please see page no. 2 marked with yellow color):

The RED fouling is fundamentally different from the electrodialysis (ED), and electrodialysis reversal (EDR) processes [23]. In general, the salinity gradient power generation was governed by monovalent ions. However, multivalent ions strongly contributed to the fouling formation of the IEMs.

- The authors also focused on chemical modification as a promising antifouling strategy. However, there are alternative physical or enzymatic methods that could also be considered. Besides, membrane cleaning also represents an useful technique to increase the lifetime of the membranes by reducing surface fouling. The reviewer would like to recommend the authors to include these topics in the discussion of the manuscript.

Response:  We look forward to the recommended methods and points. We highly appreciate the reviewers for their constructive comments. We have included a few statements in the main draft (Please see page no. 2 Marked with yellow color).

Changes made in the main manuscript (Please see page no. 2 marked with yellow color)::

Several antifouling strategies have been introduced, including physical, chemical, and biochemical processes. The physical cleaning was carried out using reverse flow, removing air bubbles, ultrasonication, and mechanically are some popular approaches. In contrast, chemical cleaning was achieved by the use of chemical reagents, like acids (sulfuric), bases (soda), and oxidants (hydrogen peroxide). The biochemical cleaning process uses bioactive agents such as the enzymatic process to clean the membrane.

- During the fabrication of the composite membrane, environmentally unfriendly solvents such as N-methyl-2-pyrrolidone (NMP), dichloromethane and dimethylacetamide (DMAc) are utilized. In the last years, the research community is seeking for developing more environmentally friendly polymers for membrane surface modification and membrane fabrication for both CEMs (10.1016/j.desal.2019.114183) and AEMs (10.3390/membranes10060134). Could the authors provide a greener alternative for future research to substitute the above-mentioned solvents or propose a more environmentally-friendly synthesis route?

Response:  Thanks for your valuable comment regarding environmentally friendly solvent for membrane surface modification and fabrication. We also understand the unfriendly solvent behavior.

We want to mention here that the solvents were not selected by ourselves. We have performed an initial literature survey. The solvents are chosen purely based on polymer solubility and membrane forming capability. The polymer selected for this work is because of its outstanding physicochemical properties. Also, it was not very well studied previously for reverse electrodialysis application. We have provided why we decided on the following solvent for membrane preparation.

In general, the polymer in the presence of CH2Cl2 and ClSO3H releases HCl and yields sulfonation of polymer, in our case S-PVDF.

N-methyl-2-pyrrolidone (NMP): Reason: In practice, very few solvents are satisfactory since a suitable solvent must be utterly inert to the reactants and must dissolve the functionalized polymer. The NMP solvent meets all these requirements and has been the solvent generally used commercially previously.

Dichloromethane: Reason: Here in this work, a common solvent was required that can work with polymer as well as for chlorosulfonic acid. We found that dichloromethane suited well for our experiment

In general, the solubility of chlorosulfonic acid is good in chloroform and dichloromethane solvents. In contrast, chlorosulfonic acid is slightly soluble in carbon disulfide and carbon tetrachloride. Please note:  Dichloromethane may be completely recycled.

Dimethylacetamide (DMAc): Reason: Acetone or dimethylacetamide are suitable solvents for PVDF-HFP. It is because the Hansen solubility parameters of these solvents are closer to those of that polymer.

Overall, It is found that water and organic alcohol-based solvents are the most environment-friendly solvent. But it is not possible to use these solvents because chlorosulfonic acid is a strong oxidizing acid and reacts violently with water and alcohol. Dangerously incompatible with combustible materials and may fail the rection.

- Since the contact angle of the prepared membrane decreased (higher hydrophilicity) after 10 min, the reviewer wonders if the authors perform any characterization after using the membrane in the RED stack. What is the final contact angle value after use? The same is applicable for the proton conductivity of the membranes and other physico-chemical properties. This will provide useful information to improve the discussion of this work.

Response:  This is an interesting comment. It could be a subject for further study because when we started writing this manuscript, we specifically wanted to understand the nanostructure of the prepared membrane and its output performance.

Regarding Contact angle: Pure PVDF contact angle dropped down to 68 o, and the S-PVDF contact angle dropped to 67o.

Characterization after using the membrane in the RED stack

Although we did not perform any characterization after the membrane was utilized in the RED stack. It is because we restrict ourselves to the analysis of as-prepared membranes. We appreciate the reviewers' suggestion regarding characterizing the used membrane, and we feel it is an interesting area for future research.

We look forward to performing such a study in our subsequent analysis with the addition of a degradation study, as the reviewer can see that the objective of this work is to synthesize alternative CEM-based hybrid membrane for power generation. Our focus is on understanding the physics behind the interaction of ionic species with hydration and how it affects proton conduction. 

Regarding proton conduction: We usually measure the temperature dependence of proton conduction. But as the prepared membrane was used at room temperature application. Therefore, we did not go in that direction. Our primary interest is to check that the metal-free functionalized nanomaterial can alter the membrane's nanostructure and its output performance. 

Overall, this manuscript not only bridges the theoretical and experimental aspects of the hybrid membrane technology but also contributes to paving a path for developing an energy device for power generation. This article will help us understand a real-world problem or provide a path for further research for our general intelligence. We hope the reviewer will understand our focused objective and will welcome our justification.

- The reviewer would like to know if the authors determine the membrane electro-resistance, which represents an essential figure of merit in RED approaches.

Response:  We have calculated the bulk resistance of each membrane from which we have determined the ionic conductivity.

…………..  (3)

The ionic conductivity of prepared membranes was measured by AC impedance spectroscopy (Gamry potentiostat) at 25±3 oC and calculated using equation (3). Before measurement, membranes were dipped into 1M H2SO4 aqueous solution for complete ionization of the sulfonic acid group.

- What is the experimental time in the RED stack assembly? How is the evolution of the power density with time for the maximum value?

Response:  The electrochemical characteristics curve of the RED stack in terms of power density was obtained by the galvanostatic method obtained by measuring the terminal voltage while varying the current step 2mA per minute.

- The model solutions that the authors utilized in the RED stack were based on NaCl (0.6 M and 0.01 M, respectively). However, it is widely known the high negative effect of multivalent ions (present in natural streams) on the generated power density (please see as relevant references: 10.1016/j.memsci.2008.12.042; 10.1016/j.memsci.2019.117385). To improve the quality of the work and to show the behavior of the prepared membranes in a more real scenario, the reviewer would like to recommend the authors to introduce representative model solutions (e.g., MgCl2, CaCl2 and Na2SO4) in the system.

Response:  We agree with the reviewer's comment that the multivalent salt solution has an adverse effect on the output power density of RED. We have chosen NaCl because it offers a high salinity gradient and excellent solubility with high ionic conductivity. Further, lower ionic radii for ions associated with NaCl than multivalent ions enhance ion transport through the membrane. Thus, it directly affects RED performance. NaCl salt is excessively available in seawater, and other salts elements are relatively deficient. Therefore it is easy to compare our result with the reported literature to get a more comprehensive picture.

The article mentioned by the reviewer is highly reputed and covers the specialized topic of multivalent and divalent ions.

The objective of the article:

https://doi.org/10.1016/j.memsci.2008.12.042: To investigate the effect of multivalent ions in the feed solutions on the power density of a reverse electrodialysis system

https://doi.org/10.1016/j.memsci.2019.117385: Modelling the effect of divalent ions on RED performance and the influence of divalent ions on cation and anion exchange membrane resistances.

Please note: The objective of the work reported in the literature is totally different from ours. Although, the application is the same, i.e., power production using RED.

Our article: Mainly deals with the polymer chemistry of membranes and focuses on low-cost hybrid membrane preparation for power production. 

- The authors report a maximum power density of 0.2 W/m2 using S-DPVDF-5 (as CEM) and a commercial Neosepta AEM at a flow rate of 100 ml/min. However, the same result is achieved when using both commercial Neosepta CEMs and AEMs. In other words, the use of the prepared CEM do not improve the performance of commercial pristine ion-exchange membranes regardless the higher proton conductive properties of the prepared composites. The authors should make a comprehensive comparison between the commercial CEM and the prepared sample (stability, physico-chemical properties, proton conduction, electro-resistance, etc.).

Response:  We agree with the review's comment regarding low power density value as compared to the Neosepta. Although we have provided a comparative table, it shows a different system and proton conductivity.

The high cost of the commercially available membrane is the major factor associated with the device. Single-handedly membrane itself is responsible for making RED technology expensive. The use of low-cost material for developing tailored-made membranes is an alternative and a practical approach for lowering the device cost and making it viable for designing a pilot plant.

Although, the output performance of the prepared membrane is similar to the commercially available membranes. But there are other factors such as swelling density, thickness, and ion exchange capacity, and the thermal stability is outstanding. The results achieved in this work are promising and are subject to further optimization, which possibly enhances the performance. The prepared hybrid membrane acts as a potential candidate for competing with commercially available membranes.

The ion conductivity of the membrane mainly depends on the ion-exchange capacity of the membrane. The higher the ion-exchange capacity, the better will be the ionic conductivity. Note: It is challenging to achieve a higher IEC value with a stable and durable free-standing membrane for an electrochemical cell. Our main objective in this work is to develop a low-cost, high-performance tailor-made hybrid ion exchange membrane for energy application. Additionally, we discussed the nanostructure of the prepared membrane in great detail, which is unusual.

Changes made in the supporting information section of the manuscript are present below:

Table S3 Summary of physicochemical properties for commercial membranes and the s-PVDF developed in this work.

Specification

s-PVDF

(This work)

Aquivion[1]

Solvay

Nafion[2] N115 DuPont

Nafion[2]

N117 DuPont

Hyflon[3] Solvay

Neosepta[4] CSE

ASTOM

Selemion[5] CMV

Asahi 

Water uptake (%)

19.57

~35

38

38

45

25

22

Ion exchange capacity (meq g-1)

0.30

1.02

0.95-1.01

0.95-1.01

0.825

1.5-1.8

1.3-2.4

Conductivity

(S cm-1)

3.64x10-3 at 23 oC

0.02 at 50 oC, 50% RH

0.1 at 23 oC, 50% RH

0.1 at 23 oC, 50% RH

~0.1

1.8 (ohm cm2) at 25oC NaCl sol

~0.006 in NH4Cl sol

Thickness (m)

~100

24.6 ±â€¯0.7

127

183

63

160

137

Solvent

N-methyl-2-pyrrolidone (NMP)

Isopropanol (IPA)

Isopropanol (IPA)

Isopropanol (IPA)

-

-

-

The authors must also provide future prospects for improving the performance of the prepared membranes in the RED system (membrane preparation, surface modification, etc.). What is the main reason to achieve the same performance in comparison with the use of the commercial Neosepta CEM? The discussion in this respect must be improved.

Response:  We are aware that any device fabrication requires optimization of various parameters such as compartment, stack alignment, electrode compatibility, flow rate, rinse solution, pressure, temperature, and other conditions, so on. They could be an interesting topic for our future work. To improve the membrane device performance, our main target will be to enhance the ionic conductivity by increasing the ionic phases and then the stability of the membrane with its output performance.

We want to mention our previous answer to the second part of the question. The RED performance mainly depends on the combination and compatibility of CEM and AEM. Suppose CEM has high ionic conductivity or high IEC value and AEM has relatively low. Then AEM will be considered as governing membrane for RED performance. Thus, direct comparison of efficiency with other reported literature may be vague or misleading in this case. Most of the commercially available membrane including Neosepta AMX/CMX (Tokuyama), FumasTech FAD/FKD (GmbH), Qianqiu AEM/CEM, (Hangzhou Qianqiu), Selemion AMV/CMV (Asahi Glass), and Fuji AEM/CEM (Fujifilm), PC-SK and PC-SA (PCCell), and Ralex CMH- PES/AMH-PES (MEGA) offer maximum power ranging from 0.33 to above 1 W m-2 (see table). However, a handful of literature is available, which talk about handmade or tailor-made membrane for RED application. A combination of PECH and sPEEK_1.76 based membrane[8] offers a power density is 1.28 W m-2 at room temperature with a swelling density of 49 % and 35.6 %, respectively. The power density of s-PVDF (hybrid membrane) and Neosepta based membrane offers 0.2 W m-2 at ambient condition, with a stable performance at varying flow rate and negligible swelling density.

- The comparison with the literature should be included to highlight the potential applicability of the novel composites proposed.

Response:  The application of s-PVDF-HFP in RED is demonstrated in this work.. Furthermore, it could be potentially utilized in other applications such as proton exchange membrane fuel cell (PEMFC) and redox flow battery application.

Changes made in the supporting information section of manuscript:

Table S4. Summary of the power density for commercially available membranesand the sPVDF/Neosepta membrane used in this work. Here, A: Active membrane area; N: Number of cell pairs; Pmax: Maximum power density.

Type

Area (cm2)

N

Pmax (W/m2)

Ref

sPVDF/Neosepta

7x7

1

0.2

This work

Neosepta CMX (Tokuyama)/Fuji T1 AEM (Fujifilm)

10x10

5

~0.2*

[9]

Ralex CMH/AMH, (MEGA)

10x10

5

∼0.26* 

[10]

Neosepta AMX/CMX (Tokuyama)

10x10

10

~0.42*

[10]

Fuji T1 CEM/Fuji T1 AEM (Fujifilm)

6.5x6.5

10

~0.7*

[11]

Neosepta ACS/CMS (Tokuyama)

10x10

5

3.8

[12]

FumasTech FAD/FKD (GmbH)

10x10

50

6.7 (60 °C)

[12]

Qianqiu AEM/CEM, (Hangzhou Qianqiu)

25x75

25

∼0.83

[13]

Selemion AMV/CMV (Asahi Glass)

10x10

5

1.18

[14]

FumaTep FKD and FAD (GmbH)

10x10

25

1.17

[14]

FumaTep FKD and FAD (GmbH)

10x10

50

0.93

[15]

Qianqiu Heterogeneous AEM/CEM (Hangzhou QianQiu)

10x10

5

1.05

[14]

Fuji AEM/CEM (Fujifilm)

10x10

25

1.06

[16]

PC-SK and PC-SA (PCCell, Germany)

10x10

10

~0.33

[17]

Neosepta AMX/CMX (Tokuyama)

100

3

0.87

[18]

Neosepta CMX/AMX (Tokuyama

6x13

5

0.59

[19]

FKS/FAS (FumaTech GmbH)

10x10

5

2.2

[20]

FKS/FAS (FumaTech GmbH)

10x10

5

0.5

[20]

Neosepta CMX/AMX (Tokuyama)

10x10

2-30

0.95

[21]

Ralex CMH- PES/AMH-PES (MEGA)

5x5

5

0.62

[22]

Ralex CMH- PES/AMH-PES (MEGA)

10x10

5

0.65

[22]

- The authors conclude that the prepared membrane shows stable performance for RED without fouling. However, any model foulant is added into the model solutions in the experiments with the RED stack (only NaCl aqueous solutions are involved). This should be clarified and justified. The conclusions must be improved, highlighting the improved characteristics of prepared samples.

Response:  We apologize for any ambiguity that the reviewer experienced while studying our manuscript. We did not use multivalent ions for salinity gradient power generation in our case. We have only used laboratory-grade NaCl salt solution for salinity gradient power generation. However, all tests were performed at a laboratory scale to check the performance of hybrid membranes with one solution. Further, to improve the research work in the future, we will definitely consider the recommendation as mentioned above. Now we have modified our statement.

Changes made in the main manuscript:

The prepared membrane and the commercially available membrane's power output performance exhibits stable performance with a slight increment 0.015 Wm-2 at varying flow rates from 60 to 100 mL min-1 without damaging the membrane.

Reviewer 3 Report

The present manuscript reminds of works on bulk modification of PFSA membranes, like the introduction of silica and zirconia nanoparticles. It is interesting that usually such approaches require balancing the conductivity and selectivity, but RED does not have the problem of selectivity loss due to back diffusion, so such modification looks especially promising here.

I'm not that sure about the economic feasibility of RED or its environmental impact, but the work presented here describes the promising modification technique that would stand on its own even without RED in consideration.

I don't have large questions, please see done small notes below.

The authors provide very thorough list of references, but since we are talking about the width of the channels, I would cite Kreuer's model (note that I'm not affiliated with the authors in any form).

Inclusion of chemical formulas would be welcome.

Line 31, «NEOSEPTA,» - Neosepta is a trademark that contains several anion exchange (and several cation exchange) membranes with different properties. I for example work with Neosepta AMX. You should list which membrane was precisely used.

Is empty line between the paragraphs included in the current template of the Membranes?

Section 2.3 – how the thickness was maintained?

Line 120, ml – I often see the recommendation to use the capital L for litres to avoid the mixup between the small l and the capital I (and, in some fonts, numeric 1).

Line 129, ph~7 – lacking capitalization due to typo, I guess?

Table 1, meqgm-1 – which measurement units is this?

Author Response

Reviewer 3

The present manuscript reminds of works on bulk modification of PFSA membranes, like the introduction of silica and zirconia nanoparticles. It is interesting that usually such approaches require balancing the conductivity and selectivity, but RED does not have the problem of selectivity loss due to back diffusion, so such modification looks especially promising here.

I'm not that sure about the economic feasibility of RED or its environmental impact, but the work presented here describes the promising modification technique that would stand on its own even without RED in consideration.

I don't have large questions, please see done small notes below.

The authors provide very thorough list of references, but since we are talking about the width of the channels, I would cite Kreuer's model (note that I'm not affiliated with the authors in any form).

Response:  We appreciate the reviewer’s suggestion and we included the reference in our revised manuscript.

Klaus-Dieter Kreuer; Prof. Dr. Albrecht Rabenau; Dr. Werner Weppner (1982). Vehicle Mechanism, A New Model for the Interpretation of the Conductivity of Fast Proton Conductors. , 21(3), 208–209. doi:10.1002/anie.198202082

Changes made in the main manuscript (Please see page no. 10 marked with yellow color):

Vehicle Mechanism is responsible for the proton conduction in the ion-exchange membrane, which promotes H3O+ ions.

Inclusion of chemical formulas would be welcome.

Response:  Figure 3 in the main manuscript represents the chemical structure of the PVDF-HFP polymer and the graphitic carbon nitride.

Regarding NEOSEPTA: Poly(styrene-divinylbenzene) copolymer containing quaternary ammonium for AEM and sulfonic group for CEM

Figure Chemical structure of the NEOSEPTA polymer backbone.

Line 31, «NEOSEPTA,» - Neosepta is a trademark that contains several anion exchange (and several cation exchange) membranes with different properties. I for example work with Neosepta AMX. You should list which membrane was precisely used.

Response:  We have used NEOSEPTA CMX as cation exchange membrane and NEOSEPTA AMX as anion exchange membrane.

Is empty line between the paragraphs included in the current template of the Membranes?

Response:  The empty space between the consecutive paragraph was removed in the revised manuscript.

Section 2.3 – how the thickness was maintained?

Response:  The film thickness was measured by using a digital micrometer.

Line 120, ml – I often see the recommendation to use the capital L for litres to avoid the mixup between the small l and the capital I (and, in some fonts, numeric 1).

Response:  Thank you for the constructive feedback. Now we have corrected the ml with mL as the reviewer suggested. (Please see page 3 marked with yellow color)

Line 129, ph~7 – lacking capitalization due to typo, I guess?

Response:  Yes, the reviewer is correct and it should be pH, not ph. We have revised it. (Please see page 3 marked with yellow color)

Table 1, meqgm-1 – which measurement units is this?

Response:  We apologize for the typo. We have revised the unit for the ion exchange capacity. It should be meq g-1

Round 2

Reviewer 1 Report

Misprint:

Line 423 (Table 1):

(mmol-eq g-1) instead of (meq gm-1). According to IUPAC recommendation [Nomenclature for chromatography (IUPAC Recommendations 1993) L. S. Ettre // Pure &Appl, Chem., Vol. 65, No. 4, pp. 81H72, 1993. https://doi.org/10.1351/pac199365040819 ], the unit of capacity is amount (mmol) of ionogenic group per mass (g) of dry ion exchanger.

Author Response

Response) We highly appreciate that the reviewer provided us a reputable reference (Nomenclature for chromatography: IUPAC Recommendations 1993) in order to improve the quality of our manuscript. However, we are confident that the unit (meq g-1) used for the ion-exchange capacity in table 1 (Line 423) is correct and globally accepted by the scientific community in the ion-exchange membrane field [Membranes, 2021, 11, 816, Scientific Reports, 2022, 12, 3686]. Also, the current notation used in our manuscript allows for a direct comparison of our IEC values with those literature values.

Please note: We noticed that the term (mmol) suggested by the reviewer is used in several places for determining the ionogenic group in the capacity of the ion exchanger in chromatography. In our case, we have calculated the ion-exchange capacity (IEC) for either ion-exchange membrane (IEM) or polyelectrolyte. With due respect, we would like to bring the reviewer's attention that the unit representation in (meq) and (mmol) are essentially identical. But, the physics behind it is a little different in both units' manifestations.

Quantitative Explanation:

In (mmol) case: Unit suggested by reviewer #1:

The molar concentration is defined as the amount of substance of a dissolved substance divided by the solvent volume. In chemistry, the commonly used unit is mmol/L = mol/dm³ = mol/m³. In case of very high concentrations of the dissolved substance also mol/L.

In (meq) case: Unit presented by the author:

The equivalent concentration is the amount of substance of a dissolved substance divided by the solvent volume multiplied by the diluted substance's valency (z).

According to SI conventions, the unit of the equivalent concentration is (mol/L ⋅ 1/z). Now, it is outdated in chemistry. Instead, the unit is represented as  (meq/L) and commonly used. In the case of high concentrations of the dissolved substances, also expressed as  eq/L.

  • For converting a molar concentration into the corresponding mass concentration, multiply the molar concentration with the molar mass (M).

Molar concentration  (mmol L-1) = (M) Mass concentration (mg L-1)

(Here: M = molar mass)

  • Likewise, for converting a molar concentration into the corresponding equivalent concentration, multiply the molar concentration with the valency (z).

Molar concentration  (mmol L-1) = (z) Equivalent concentration (meq L-1)

(Here: z = valence)

Thus, mmol L-1 = (z) meq L-1  or similarly,  mmol g-1 = (z) meq g-1

(Note: here, (M) and (z) are the conversion factors of different concentration units)

Reviewer 2 Report

The reviewer would like to thank the authors for the detailed and specific responses provided. The quality of the manuscript has been improved.

Author Response

We thank the reviewer #2 for his/her positive comment.
